# Retinoid Regulation of Ocular Surface Innate Inflammation

**DOI:** 10.3390/ijms22031092

**Published:** 2021-01-22

**Authors:** Jehan Alam, Zhiyuan Yu, Cintia S. de Paiva, Stephen C. Pflugfelder

**Affiliations:** Department of Ophthalmology, Baylor College of Medicine, Houston, TX 77030, USA or Jehan.Alam@bcm.edu (J.A.); Zhiyuan.Yu@bcm.edu (Z.Y.); cintiadp@bcm.edu (C.S.d.P.)

**Keywords:** retinol, retinoic acid, cornea, conjunctiva, dry eye, inflammation, NFκB

## Abstract

Corneal and conjunctival inflammation and dry eye develop in systemic vitamin A deficiency (VAD). The objective of this study was to investigate the lacrimal ocular surface retinoid axis, particularly immunomodulatory effects of retinoic acid (RA) and change in conjunctival myeloid cell number and phenotype in VAD. We discovered that ocular surface epithelial and myeloid cells express retinoid receptors. Both all *trans*- and 9-*cis*-RA suppressed production of dry eye relevant inflammatory mediators [interleukin(IL)-1β, IL-12, regulated upon activation, normal T cell expressed and secreted (RANTES)] by myeloid cells. Systemic VAD was associated with significant goblet cell loss and an increased number of CD45+ immune cells in the conjunctiva. MHCII^−^CD11b^+^ classical monocytes were significantly increased in the conjunctiva of VAD C57BL/6 and RXR-α mutated Pinkie strains. RNA seq revealed significantly increased expression of innate immune/inflammatory genes in the Pinkie conjunctiva. These findings indicate that retinoids are essential for maintaining a healthy, well-lubricated ocular surface and have immunomodulatory effects in the conjunctiva that are mediated in part via RXR-α signaling. Perturbation of the homeostatic retinoid axis could potentiate inflammation on the ocular surface.

## 1. Introduction

Knowledge regarding the pathogenesis of the ocular surface disease of vitamin A deficiency continues to evolve. In ancient times, Egyptian physicians found that treatment of night blindness with liver from an ass or ox improved this condition, and the nutritional component in the liver responsible for this was eventually identified as vitamin A. In 1816, François Magendie, a pioneering physiologist, found that dogs fed a diet containing sugar in distilled water without animal products lost weight, developed corneal ulcers, and subsequently died [1]. Charles-Michel Billard, a French pediatrician, reported in 1828 that children with complete marasmus develop softening of the cornea, with an appearance similar to what was observed in the malnourished dogs [1]. Lunin (1881) observed that mice had a greater survival rate of when whole dried milk was added to their diet [2]. In the early 1900s, E. McCollum at the University of Wisconsin and Osborne and Mendel at Yale independently discovered that young rats fed a basic diet consisting of carbohydrate and protein sources had better growth and survival when their diet was supplemented with butter fats or extract either of butter fats, egg yolk, and/or liver. The unknown factor in these supplements was called “fat-soluble factor A”. Apart from defective growth and survival, the young rats fed the basic diet developed eye disease, which could be alleviated by adding butter fats [1,2]. Thus, by 1917, it was experimentally established that lack of a fat-soluble factor in the rat diet caused weight loss and xerophthalmia that could result in blindness. Subsequently, in 1931, the swiss chemist Paul Karrer extracted vitamin A from cod-liver-oil and described its chemical structure, and vitamin A was eventually chemically synthesized by Isler and his group in 1947 [2].

It is now known that vitamin A is a group of nutritional unsaturated organic compounds that consist of an unsubstituted β-ionone ring and isoprenoid chain. These include alcohol (retinol), aldehyde (retinal) and ester (retinyl ester) forms, retinoic acid, and several provitamin A carotenoids—mostly notably the β-carotene.

Humans ingest vitamin A either as retinyl ester found in animal sources, including eggs, liver, fish, and dairy products, or in the carotenoid forms, α-carotene, β-carotene, and β-cryptoxanthin, from plants. During intestinal digestion, retinyl ester and provitamin A carotenoids are partially or completely hydrolyzed into free retinol, which enters enterocytes either by simple diffusion or a carrier-mediated process [3]. Once inside enterocytes, retinol is re-esterified with long chain fatty acids, such as palmitate, and incorporated along with other dietary lipids into triacylglycerol-rich chylomicrons that are released into the lymphatics, while free retinol is also released into the portal circulation [4]. In the blood, the triacylglycerol in chylomicrons is hydrolyzed by lipoprotein lipase, resulting in the production of “chylomicron remnants”, which are exclusively taken up by hepatocytes and stored predominantly as retinyl ester in the perisinusoidal stellate cells. Approximately 70% of the vitamin A in the body is stored in these hepatic stellate cells [5]. Prior to release from the liver, retinyl ester is hydrolyzed into free retinol and complexed with retinol binding protein (RBP) [6]. Plasma RBP is the high affinity specific retinol carrier protein in the blood that stabilizes retinol and delivers it to the target tissues [7]. RBP, either through a cell-specific receptor or facilitated diffusion, delivers retinol to the lacrimal gland acinar cells, where it is stored as fatty acyl esters (retinyl linoleate, retinyl palmitate, retinyl stearate) [8,9,10]. Retinol bound to RBP is secreted by the lacrimal gland into the tears [10]. Retinol concentration in secreted lacrimal fluid was reported to fall to non-detectable levels in vitamin A deficient rabbits [10]. Cellular and biochemical changes were observed in lacrimal gland acini of vitamin A deficient rats, including decreased secretory granules, endoplasmic reticulum atrophy, and reduction of histochemically detectable glycoproteins [11].

These findings indicate that vitamin A is essential for maintaining a lubricated uninflamed ocular surface and clear cornea. While the vitamin A deficient phenotype is well characterized, the biologic activity of retinoids on the ocular surface is incompletely understood. The objective of this study was to investigate the lacrimal ocular surface retinoid axis, particularly the immunomodulatory effects of retinoic acid (RA) and change in conjunctival myeloid cell number and phenotype in VAD.

## 2. Results and Discussion

### 2.1. Lacrimal Glands Secrete Vitamin A into Tears that Is Metabolized on the Ocular Surface

We found mouse corneal and conjunctival epithelia express RBP4 and its specific cellular receptor, stimulated by retinoic acid 6 (*STRA6*) (Figure 1A). Interestingly, expression of both of these receptors increased in the conjunctiva of mice exposed to 10 days of experimental desiccating stress (DS10) induced dry eye (Figure 1B). Expression of RPB4 has been recognized to increase in a number of chronic inflammatory diseases and in oxidative stress [12].

The ocular surface epithelial cells (particularly the conjunctival goblet cells) also express alcohol and aldehyde dehydrogenase (aldh1a1 and a3) enzymes that metabolize retinol to the biologically active form of vitamin A, retinoic acid (RA), which exists in vivo primarily as all-*trans*-retinoic acid (ATRA) and its isomer 9-*cis* RA [13]. The Th1 cytokine, IFN-γ, which increases in dry eye and causes goblet cell secretory dysfunction and apoptosis significantly reduced levels of aldh1a3 transcripts (1.65 log fold) when added to cultured goblet cells [14]. In contrast, the aldh1a2 isoform is expressed by myeloid phagocytic cells [13], and both epithelial and myeloid cells contribute to aldehyde dehydrogenase activity on the ocular surface, which was found to decrease with aging [15]. Both aging and dry eye are also associated with antigen-presenting cell activation in the conjunctiva [15,16], and we found that, similar to aging [15], expression of aldh1a2 in the conjunctival significantly decreased after 5 days of experimental desiccating stress (DS) induced dry eye (Figure 1C).

These findings indicate the ocular surface is a retinoid-rich environment and that retinol secreted by the lacrimal gland can be metabolized into retinoic acid by epithelial and myleloid cells residing in the ocular surface tissues. Levels of aldehyde dehydrogase expression and activity decrease with aging and dry eye and reduced retinoid conditioning may contribute to the increased antigen presenting cell activation and ocular surface inflammation that is found in these conditions.

### 2.2. Retinoic Acid Receptor Expression on the Ocular Surface

Retinoic acid receptors are expressed by the epithelial and myeloid cells on the ocular surface. Bossenbroek et al. found retinoid acid receptors (RARs) α, β, and γ and retinoid X receptors (RXRs) α and β gene transcripts are expressed by conjunctival fibroblasts and corneal epithelium and stromal cells. Mori and colleagues immunodetected RARα and γ and RXR α and β in the adult mouse cornea (epithelium and stroma) and conjunctiva [17]. Suzuki et al. found 1.8-fold (*p* < 0.05) higher expression of RXRα in human cultured cornea epithelium of females compared with males [18]. Our group reported that 49% of MHCII^−^ and 85% of MHCII^+^CD11b^+^ myeloid cells in the conjunctiva express RXRα, and this was also confirmed in CD11b^+^ cells by imaging flow cytometry (Figure 2A) [19]. These findings indicate the epithelial and myeloid immune cells on the ocular surface express retinoid receptors and are potentially responsive to retinoids.

### 2.3. Effects of Retinoic Acid on Innate Immune Cells

NFκB is activated in the mouse corneal epithelium in response to experimentally induced desiccation stress (Figure 2B), and stimulates production of NFκB inducible cytokines and chemokines, including IL-1β, IL-12, and RANTES [20]. We found that both synthetic and conjunctival goblet cell derived retinoic acid (RA) suppressed production of IL-12 by LPS stimulated cultured bone marrow derived myeloid cells and in vivo by CD11b^+^ cells in the mouse conjunctiva [13,21]. We also found that both 9-*cis* and all *trans*-RA suppress the production of dry eye associated inflammatory factors (IL-1β, IL-12, RANTES) and increase the production of the anti-inflammatory cytokine IL-10 by LPS-stimulated bone marrow derived monocytes (Figure 2C).

We also evaluated the effects of systemic vitamin A deficiency (VAD) and the loss of function RXRα mutation in the Pinkie mouse strain on goblet cell density and immune cell populations in the conjunctiva. VAD in mice was created by a previously reported protocol [22], and confirmed by reduced expression of stimulated by retinoic acid inducible gene, Stra6 protein, and mRNA in the corneal epithelium (Figure 3A) and a significant reduction in the number of conjunctival goblet cells (Figure 3B).

Because we have found an increase in CD11b^+^ myeloid cells in the conjunctiva of the SPDEF^−/−^ mouse strain that lacks retinoic acid-producing goblet cells [21], we compared percentages of CD45^+^ and myeloid (MHCII^−^CD11b^+^) cells in the conjunctiva of C57BL/6 fed a normal vitamin A containing control or VAD diet and in the Pinkie strain by flow cytometry. There was a significant increase in the number of CD45^+^ and MHCII^−^CD11b^+^ cells in the conjunctival of both VAD and Pinkie groups (Figure 4A,B) and these were identified as Ly6C^high^CD43^lo^ classical monocytes in Pinkie (Figure 4D,E).

Similar to VAD mice, the Pinkie strain also has goblet cell loss (manuscript in preparation) and has been reported to develop dry eye [22]. To gain insight into the role of RXRα signaling on maintaining ocular surface homeostasis, we performed RNAseq on the whole conjunctiva to compare the gene expression profile between Pinkie and the wild type C57BL6 mice. We found that, out of 18,909 genes, there were 1375 with significant differential expression (Figure 5A) These genes included those associated with epithelial differentiation, monocytes/macrophages, inflammatory signaling, cytokines/chemokines, proteases, and MHCII and IFN-γ inducible genes (Figure 5B).

These findings indicate that VAD causes myeloid cell infiltration of the ocular surface, and similar findings are found in mice with reduced RXRα signaling. RXRα regulates innate inflammation in the conjunctiva and reduced RXRα signaling may contribute to the ocular surface inflammation and dry eye that develops in systemic VAD. They also indicate that RXRα ligands are potential therapeutic targets to suppress disease causing inflammatory cells and mediators in dry eye.

## 3. Conclusions

The lacrimal gland secretes the retinol form of vitamin A in tears, where it is taken up by ocular surface epithelial and myeloid cells and metabolized into retinoic acid (RA). Based on the consequences of systemic vitamin A deficiency and reduced RXRα signaling, there appears to be a retinoid axis on the ocular surface that is required for maintaining a healthy, well lubricated ocular surface and suppressing innate inflammation. While much has been learned about the mechanisms by which retinoids regulate inflammation, further investigations are needed to fully understand the function of retinoids on the ocular surface and their therapeutic potential for treating ocular surface disease.

## 4. Materials and Methods

### 4.1. Animals and Vitamin A Deficiency Protocol

The animal protocol for this study was designed according to the Association for Research in Vision and Ophthalmology (ARVO) Statement for the use of Animals in Ophthalmic and Vision Research and was approved by the Institutional Animal Care and Use Committees at Baylor College of Medicine (AN2032, approved 30 July 2019). Female C57BL/6J (B6) mice aged 6–8 weeks were purchased from Jackson Laboratories (Bar Harbor, ME, USA) and allowed to rest in a humidified environment for a week before the experiment. Vitamin A deficiency was induced with a standard protocol [22]. Briefly, beginning at 2 weeks of gestation, all pregnant females were fed a vitamin A-deficient green colored diet (58M1/NO Vit. A green); after birth, the dams either stayed on the A-deficient diet (green color) or were placed on a control yellow colored vitamin A supplemented diet (58m1 w/yellow dye). RXRα mutant (I273N) *Pinkie* mice were bred internally and maintained in a specific-pathogen-free vivarium.

### 4.2. Measurement of Goblet Cell Density

Goblet cell density was measured as previously described [23]; briefly, following euthanasia, eyes were excised (*n* = 9/group) and fixed in 10% formalin. Paraffin embedded sections, 5 µm thick, were cut with a microtome (Microm HM 340E; Thermofisher Wilmington, DE, USA). Sections were stained with periodic acid Schiff (PAS) reagent and were examined and photographed with a microscope (Eclipse E400; Nikon, Garden City, NY, USA) equipped with a digital camera (DXM1200; Nikon). Using the NIS Elements software, goblet cells were manually counted. The length of the conjunctival goblet cell zone was measured by drawing a digital line on the surface of the conjunctiva image from the first to the last PAS^+^ goblet cell. The results are presented as PAS^+^ goblet cells/mm.

### 4.3. Immunofluorescent Staining

Immunofluorescent staining was performed to detect STRA6 (Cat# 22001-1-AP, 1:100 dilution, Proteintech Group, Rosemont, IL, USA) and RBP4 (Cat# 11774-1-AP, 1:100 dilution, Proteintech Group, Rosemont, IL, USA) on the ocular surface of B6 mice. Optimal cutting temperature (OCT) embedded frozen eye tissue sections were methanol-fixed for 5 min, washed, and permeabilized with 0.3% Triton X-100 in phosphate buffered saline (PBS). Nonspecific binding sites were blocked with 20% goat serum for 1 h and incubated with primary antibodies overnight and secondary goat anti-rabbit Alexa-Fluor 488/555 conjugated IgG antibodies for 1 h. Images were captured with a confocal microscope (Nikon AR-1, Garden City, NY, USA).

### 4.4. Real-Time PCR

Following euthanasia, the corneal epithelium was scraped, and total RNA was extracted using a QIAGEN RNeasy Plus Micro RNA isolation kit (Qiagen, Valencia, CA, USA) according to the manufacturer’s instruction. The cDNA was synthesized using the Ready-To-Go^−^You-Prime-First-Strand kit (GE Healthcare, Pittsburgh, PA, USA). Quantitative real-time PCR was performed with specific Taqman probes (Life Technologies, Grand Island, NY, USA) for STRA6 (Mm00486457_m1) and RBP-4 (Mm00803264_g1). The *HPRT-1* gene was used as an endogenous reference for each reaction. The results of real-time PCR were analyzed by the comparative CT method.

### 4.5. Western Blot

Corneal epithelial cells from B6 and vitamin A deficient mouse were scrapped and placed in cell lysis buffer (Part no. 895347, R&D Minneapolis, MN, USA). Protein concentration was determined using a micro bicinchronic acid (BCA) protein assay (Cat# 23235, Thermo Fisher, Waltham, MA, USA). Here, 50 µg of corneal extract was resuspended in sodium-dodecyl sulphate (SDS) sample buffer, boiled for 5 min, and analyzed on 4–15% mini-protean TGX™ stain-free gels (Cat# 4568084, Bio-Rad, Hercules, CA, USA). The proteins were transferred to polyvinylidene difluoride membranes (Bio-Rad, Cat# 170-4157). The blots were incubated with an anti-STRA6 (Cat# 22001-1-AP, 1:100 dilution, Proteintech Group, Rosemont, IL, USA) or an anti-β actin antibody (Sigma Aldrich, St. Louis, MO, USA, Cat# A5441) overnight. After secondary antibody incubation, the signals from antigen–antibodies complexes were developed with ECL plus Western Blotting Detection kit (Cat# RPN2106, ECL, GE Healthcare, Chicago, IL, USA). Images were taken using ChemiDoc Touch Imaging System (ChemiDoc Touch Imaging System; Bio-Rad, Hercules, CA, USA), and band densities were measured by Bio-rad software (Image lab, v. 6.0; Bio-Rad).

### 4.6. Flow Cytometry

Conjunctivae from B6, vitamin A deficient, or Pinkie strains were excised, chopped with scissors into tiny pieces, and incubated with 0.1% type IV Collagenase for 1 h to yield single cell suspensions. Samples were incubated with anti-CD16/32 for 5 min at room temperature and subsequently stained with CD45 (clone 30-F11, Biolegend, San Diego, CA, USA), MHC II (clone I-A/I-E, BD Pharmingen; San Diego, CA, USA), CD11b (clone M1/70, Thermofisher Scientific, Waltham, MA, USA). Cells were stained with an infra-red fluorescent viability dye (Life Technologies, Grand Island, NY, USA). The gating strategy was as follows: lymphocytes were identified by forward -scatter area (FSC-A) and side scatter area (SSC-A) gates, followed by two singlets gates (FSC height vs. FSC-A and SSC-height vs. SSC-A), followed by live/dead identification using the infra-red fluorescent viability dye. Alive CD45^+^ cells were plotted for MHC II and CD11b expression. Negative controls consisted of fluorescence minus one splenocytes. Cells were acquired with either BD LSR II or BD Canto II Benchtop cytometers with BD Diva software v. 6.7 (BD Biosciences, San Diego, CA, USA), and 200,000 or more events were collected. Final data were analyzed using FlowJo software version 10 (Tree Star Inc., Ashland, OR, USA).

### 4.7. Measurement of NFκB p65 Activation

Corneal epithelium was scrapped using a dulled blade and nuclear protein extraction was performed according to the manufacturer’s instructions. NFκB p65 activation was measured by a TransAM NFκB p65 kit that specifically quantifies phosphorylated NFκB p65 (Cat# 40596, Active Motif, Carlsbad, CA, USA). Nuclear extracts from B6 mice maintained in a normal environment and from B6 mice with desiccating stress induced dry eye were added to wells of a 96-well plate with immobilized oligonucleotide containing an NFκB consensus binding site. The activated p65 in the nuclear extract binds to the oligonucleotide. After incubation with specific anti-p-p65 antibodies, horseradish peroxidase (HRP) conjugated secondary antibodies provided a sensitive colorimetric readout at 450 nm using a colorimetric plate reader (Tecan Infinite M200, Magellan v. 6.55 software; Tecan, Männedorf, Switzerland).

### 4.8. RNA Seq Data Analysis

Conjunctival epithelium was excised from B6 and Pinkie strains and total RNA was extracted using a QIAGEN RNeasy Plus Micro RNA isolation kit (Qiagen, Valencia, CA, USA) according to the manufacturer’s instructions. The concentration and purity of RNA were assessed using a NanoDrop 1000 (ThermoFisher Scientific, Waltham, MA, USA). RNA-Seq was performed by the Beijing Genomics Institute (BGI) using the BGISEQ500RS to generate 100 bp paired-end reads. The raw data were cleaned by removing reads containing adapter or poly-N sequences, and reads of low quality using SOAPnuke (v. 1.5.2, parameters: −l 15 −q 0.2 −n 0.05), and the expression levels of the resulting genes and transcripts were determined using RSEM (v. 2.2.5, default parameters). Detection of DEGs (differentially expressed genes) was performed with DEseq2 (parameters: fold change ≥ 2.00 and adjusted *p*-value ≤ 0.05). A total of 19,511 genes were obtained as raw data. Genes were passed through the Benjamini–Hochberg procedure to obtain the critical value for false discovery and a total of 1375 genes passed with a *p*-value > 0.0006. The selected genes were clustered in a heat map based on GSCA pathways.

### 4.9. Statistical Analysis

Based on normality, parametric student T or nonparametric Mann–Whitney U tests were performed for statistical comparisons with an alpha of 0.05 using GraphPad Prism 9.0 software (GraphPad Software, Inc., San Diego, CA, USA).

## Figures and Tables

**Figure 1 ijms-22-01092-f001:**
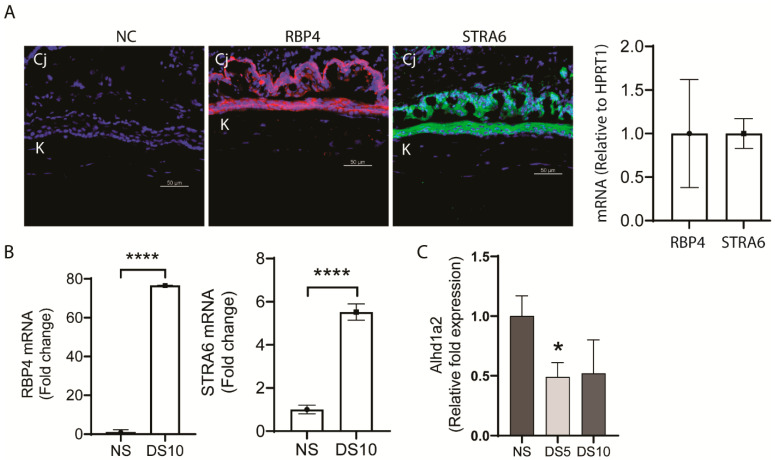
Expression of stimulated by retinoic acid 6 (STRA6) and retinol binding protein 4 (RBP4) in the ocular surface epithelium. (**A**) Representative immunofluorescence images of ocular surface epithelium in sections taken from young C57BL/6 (B6) mice showing expression of STRA6 (green) and RBP-4 (red) both in the cornea (K) and conjunctiva (Cj). The nuclei were stained with 4’,6’-diamino-2-phenylindole (DAPI) (blue) (left). Relative expression of STRA6 and RBP-4 mRNA in scaped cornea epithelium from young B6 by real time PCR (RT-PCR) (right). (**B**) RBP4 and STRA6 mRNA expression measured by RT-PCR in conjunctiva of 6-week-old B6 mice in normal non-stressed environmental conditions (NS) and after 10 days of desiccating stress induced dry eye (DS10). (**C**) Desiccating stressed induced decrease in aldh1a2 gene expression measured by RT-PCR in the conjunctiva of B6 mice. NS = control non stressed, DS5 = exposure to desiccating stress for 5 days, DS10 = exposure to desiccating stress for 10 days. * *p* = 0.05, **** *p =* 0.00001.

**Figure 2 ijms-22-01092-f002:**
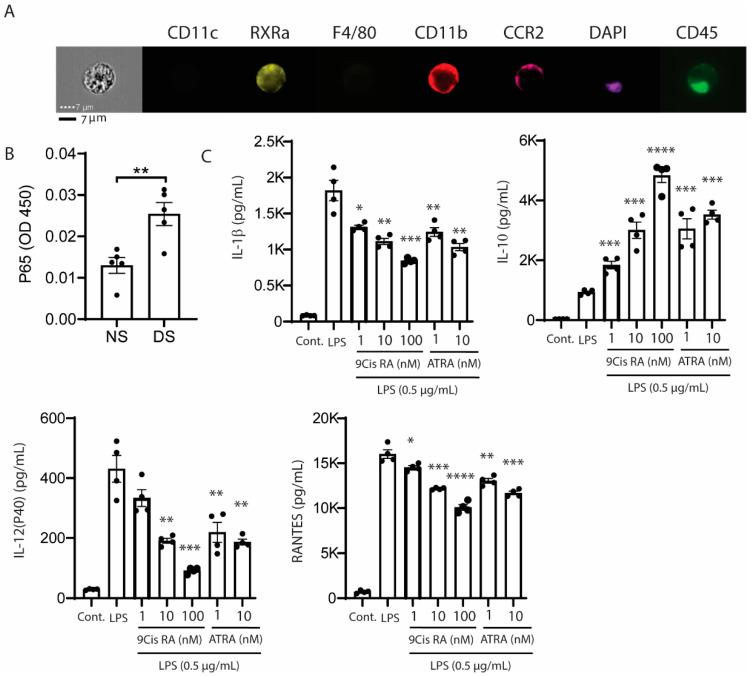
(**A**) Imaging flow cytometry showing retinoid X receptor α (RXRα) positivity in isolated CCR2+CD11b+ conjunctival myeloid cells. (**B**) Nuclear phospho-NFκB p65 increases in the cornea of B6 mice exposed to desiccating stress. NFκB activation (phospho p65) measured in nuclear fractions of corneal epithelial samples obtained from non-stressed B6 control (NS) or C57BL/6 mice following exposure to desiccating stress for 1 day (DS) using a TransAM NFκB p65 kit (Active Motif, Carlsbad, CA, USA), *n* = 5/group. ** *p* = 0.006. (**C**) Effects of 9-*cis*- and all *trans* retinoic acid (ATRA) on supernatant concentrations of interleukin( IL)-1β, IL-10, IL-12p40, and RANTES in LPS stimulated cultured bone marrow derived murine monocytes. LPS = lipopolysaccharide, RANTES = regulated upon activation, normal T-cell expressed, and presumably secreted; * *p* = 0.01, ** *p* = 0.002, *** *p* = 0.0001, **** *p* = 0.00001.

**Figure 3 ijms-22-01092-f003:**
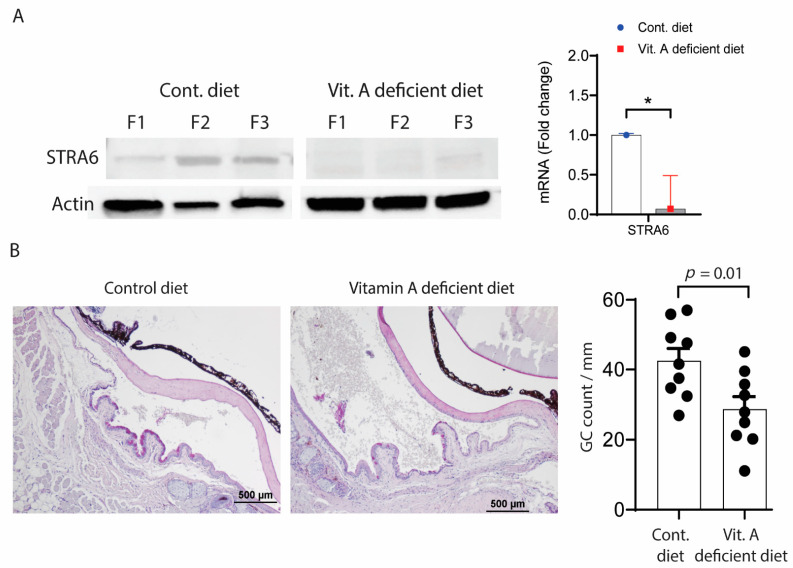
(**A**) Stimulated by retinoic acid 6 (STRA6) expression. Left. STRA6 in the cornea epithelium is downregulated with vitamin A deficiency. Western blot for STRA6 and β-actin in the cornea epithelium of C57BL/6 mice fed a normal vitamin A containing (control) or vitamin A deficient diet from three different biological samples from each group. Right. Relative fold expression of STRA6 mRNA in the conjunctiva of B6 mice on control or vitamin A deficient diet. (**B**) Vitamin A deficiency causes conjunctival goblet cell loss. Left. Representative images of filled goblet cells (GCs) counted in periodic acid Schiff (PAS) stained sections in the GC-rich zone of the conjunctiva of C57BL/6 mice fed a control or vitamin A deficient diet. Right. Each graph bar represents mean + standard deviation of goblet cells/mm. Individual dots represent average of right and left eyes per mouse (*n* = 9 per group). * *p* = 0.01.

**Figure 4 ijms-22-01092-f004:**
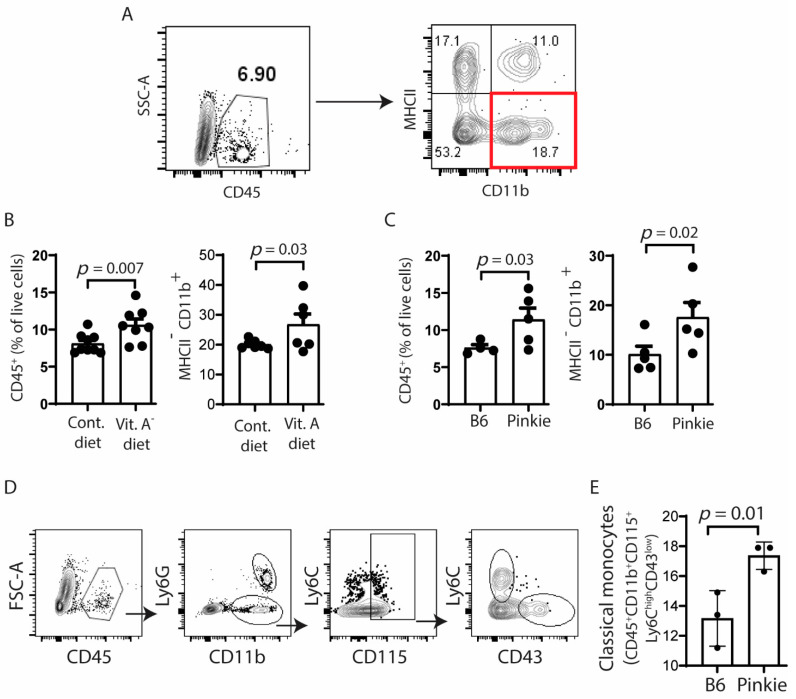
Recruitment of myeloid immune cells to the conjunctiva increases in vitamin A deficiency or attenuated RXR-α signaling. (**A**) Gating strategy for single cell suspension from conjunctiva of C57BL/6 (B6) mice on normal vitamin A containing control (Cont.) or vitamin A deficient (Vit A^−^) diets or from the RXR-α mutant Pinkie strain with attenuated RXR-α signaling stained with antibodies for CD45, CD11b, and MHCII. (**B**,**C**) Bar graphs showing the percentages of positive cells (*n* = 8/group) with each dot representing one animal; error bars indicate the standard error of mean (SEM). (**D**) Gating strategy for classical monocytes in single cell suspensions obtained from conjunctivae of C57BL/6 (B6) mice or Pinkie strains stained with antibodies for CD45, CD11b, Ly6G, Ly6C, CD115, and CD43. (**E**) Bar graph showing the percentages of classical monocytes (CD45^+^CD11b^+^CD115^+^Ly6C^hi^CD43^lo^) (*n* = 3/group) with each dot representing one animal; error bars indicate the standard error of mean (SEM). SSC, side scatter area.

**Figure 5 ijms-22-01092-f005:**
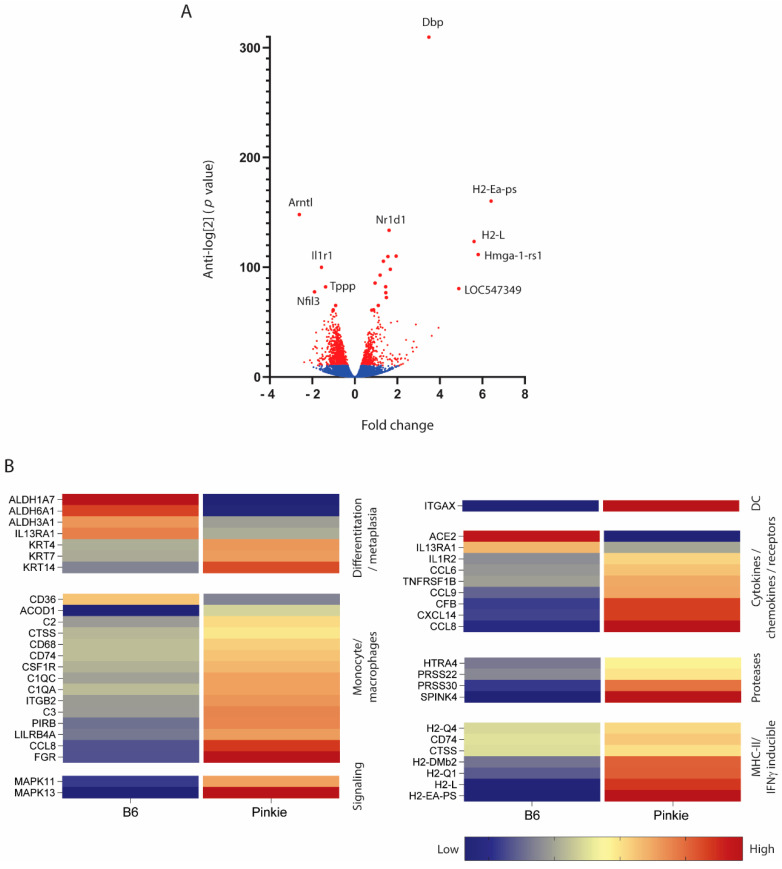
Differential gene expression from bulk RNA seq performed on C57BL/6 (B6) and Pinkie RXR-α mutant strains. (**A**) Volcano plot showing the up- and down-regulated genes in the conjunctiva of Pinkie compared with B6 control. Genes not reaching a significant difference and that did not pass the Benjamini–Hochberg procedure are shown in blue. Genes with significant between group difference are shown in red. (**B**) Heatmap of differentially expressed genes clustered into groups based on Gene Set Co-expression Analysis (GSCA) pathways. All the genes passed the Benjamini–Hochberg procedure to exclude false discovery; the selected genes had a *p*-value > 0.0006. Each row represents a specific gene: the right column represents the Pinkie strain and left column represents the B6 strain. Red colors indicate higher expression while blue colors indicate lower expression.

## Data Availability

Data is contained within the article.

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
