# Peer review of "Retinoid Regulation of Ocular Surface Innate Inflammation"

_ijms, 2021, doi:10.3390/ijms22031092_

Round 1

Reviewer 1 Report

In this manuscript, Alam et al.  have shown possible regulation of innate ocular surface inflammation by retinoid using vitamin A deficiency (VAD) animal model. They showed that there was significant increase in inflammatory cells, especially MHCII-CD11b+ cells in VAD mouse and RXR mutated Pinkie stains, and their RNA seq results supported their ideas. Overall, the manuscript is well written, and the results may be of readers' interest since authors specifically showed linkage between VAD and innate ocular surface inflammation. However, following points should be revised.

Abstract

  1. "V C57BL/6", what does author mean by that? Is it VAD?

Introduction

  1. There is too much historical information give in introduction. It would be better if authors could simplify these information. Moreover, authors should clarify their purpose of conducting the experiments regarding retinoid and ocular surface inflammation.

Results and discussion

  1. First paragraph, it would be better for readers to understand the mechanism of retinol absorption and its secretion through lacrimal gland if the given information placed in introduction.
  2. Figure 1. was there any differences in RBP4 and STRA4 expression in dry eye animal model in immunostaining? Representative images for RBP4 and STRA4 from dry eye model could give more information to readers.
  3. Page 2, line 74. RPB should be revised to RBP.
  4. Page 2, line 87. "cultured" is written twice. Please delete one.
  5. Page 3, line 107-108. retinoic acid receptor or retinoid acid receptor. Which one is correct? Authors should clarify.
  6. Page 6, line 154. "C57BL/6 fed a control". It would be better to indicate what is "a control" in the Results, although it is stated in the Methods and materials.

Materials and Methods

1. Author should clarify how statistical analysis was performed for the results displayed in the paper.

Author Response

Response to Reviewer 1

We appreciate the complementary review and suggestions for improving the manuscript.

Comments and Suggestions for Authors

In this manuscript, Alam et al.  have shown possible regulation of innate ocular surface inflammation by retinoid using vitamin A deficiency (VAD) animal model. They showed that there was significant increase in inflammatory cells, especially MHCII-CD11b+ cells in VAD mouse and RXR mutated Pinkie stains, and their RNA seq results supported their ideas. Overall, the manuscript is well written, and the results may be of readers' interest since authors specifically showed linkage between VAD and innate ocular surface inflammation. However, following points should be revised.

Abstract

  1. "V C57BL/6", what does author mean by that? Is it VAD?

Response: we apologize for the error, it should be VAD. This was corrected.

Introduction

  1. There is too much historical information give in introduction. It would be better if authors could simplify these information. Moreover, authors should clarify their purpose of conducting the experiments regarding retinoid and ocular surface inflammation.

Response: The introduction has been revised and condensed. The objective of the study is now included in the Abstract and the last sentence of the Introduction.

Results and discussion

  1. First paragraph, it would be better for readers to understand the mechanism of retinol absorption and its secretion through lacrimal gland if the given information placed in introduction.

Response: The lacrimal gland section was moved to the Introduction

  1. Figure 1. was there any differences in RBP4 and STRA4 expression in dry eye animal model in immunostaining? Representative images for RBP4 and STRA4 from dry eye model could give more information to readers.

Response: We did not obtain histological sections from the mouse dry eye model for this study, but we had conjunctival RNA from previous experiments and performed RT-PCR for RBP4 and STRA6. These results are included in paragraph 1 of the results and Figure 1B. Interestingly, expression of both of these receptors increased in the conjunctiva of mice exposed to 10 days of experimental desiccating stress (DS10) induced dry eye and expression of RPB4 has been recognized to increase in a number of chronic inflammatory diseases and in oxidative stress.

  1. Page 2, line 74. RPB should be revised to RBP.

Response: This change was made.

  1. Page 2, line 87. "cultured" is written twice. Please delete one.

The second culture was deleted.

  1. Page 3, line 107-108. retinoic acid receptor or retinoid acid receptor. Which one is correct? Authors should clarify.

Response: Retinoic acid receptor is correct and this term is used throughout the manuscript.

  1. Page 6, line 154. "C57BL/6 fed a control". It would be better to indicate what is "a control" in the Results, although it is stated in the Methods and materials.

Response: we have indicated this is a normal vitamin A containing control diet.

Materials and Methods

  1. Author should clarify how statistical analysis was performed for the results displayed in the paper.

Reviewer 2 Report

This study investigated the retinoid pathway at the ocular surface of animal models of VAD. The manuscript is well written and the topic is interesting for the readers. 

In the Abstract section please better clarify the objective of the study and highligth the results of the study rather than summarize previous findings. 

In the manuscript please state the objectives of the study and the rationale for this research both in terms of increasing understanding on pathogenic mechanisms of the ocular surface diseases in VAD and of potential therapeutic target. It should allow to better understand the results section that is quite confusing for the reader.

Author Response

Response to Reviewer 2

We appreciate the complementary review and suggestions for improving the manuscript.

This study investigated the retinoid pathway at the ocular surface of animal models of VAD. The manuscript is well written and the topic is interesting for the readers. 

Response: We appreciate the complementary review and suggestions for improving the manuscript.

In the Abstract section please better clarify the objective of the study and highligth the results of the study rather than summarize previous findings. 

Response: The objective of the study and the highlights of the results are now included in the Abstract. The summary of the previous results was removed.

In the manuscript please state the objectives of the study and the rationale for this research both in terms of increasing understanding on pathogenic mechanisms of the ocular surface diseases in VAD and of potential therapeutic target. It should allow to better understand the results section that is quite confusing for the reader.

Response: The objective of the study is now stated in the Abstract and the last sentence of the Introduction. The Results section has been revised to indicate how these findings increase understanding of the pathogenic mechanisms of the ocular surface disease of dry eye and that much of the innate inflammation may be attributed to reduced RXRa signaling and that RXRa ligands are potential therapeutic targets to suppress disease causing inflammatory cells and mediators in dry eye.